# The Challenge of Medication-Induced Dry Mouth in Residential Aged Care

**DOI:** 10.3390/pharmacy9040162

**Published:** 2021-10-01

**Authors:** William Murray Thomson, Moira B. Smith, Catherine Anna Ferguson, Geraldine Moses

**Affiliations:** 1Sir John Walsh Research Institute, Faculty of Dentistry, University of Otago, Dunedin 9054, New Zealand; 2Department of Public Health, Wellington School of Medicine, Faculty of Medicine, University of Otago, Wellington 6021, New Zealand; moira.smith@otago.ac.nz (M.B.S.); anna.ferguson@otago.ac.nz (C.A.F.); 3School of Pharmacy, University of Queensland, Brisbane 4102, Australia; g.moses@uq.edu.au

**Keywords:** xerostomia, dry mouth, polypharmacy, residential aged care

## Abstract

With a reported prevalence between 20% and 30%, dry mouth is more common among older people than any other age group. The major risk factor for dry mouth is polypharmacy. Older people take more medications than any other age group, not only for symptomatic relief of various age-associated chronic diseases, but also to reduce the likelihood of the complications that may arise from those conditions. Most aged care residents take even more medications than older people who are living in their own homes. The greater the number of medications taken, the greater the associated anticholinergic burden, and the more likely it is that the individual will suffer from dry mouth. The condition not only affects the dentition and ability to wear dentures, but also the sufferers’ quality of life. Treating dry mouth is a considerable challenge for clinicians. As medication use is by far the most important risk factor, there is a need for pharmacists, doctors and dentists to work together to prevent this from occurring. Medication review and deprescribing is a key strategy, but there have not yet been any randomised control trials of its efficacy in reducing the occurrence of dry mouth.

## 1. Introduction

Dry mouth is particularly common in residential aged care, reported in about one in three residents in a recent New Zealand (NZ) national survey [1]. The aim of this paper is to raise pharmacists’ awareness of the problem and to suggest ways in which they can work with dental personnel to ameliorate this.

## 2. Occurrence and Impact of Dry Mouth

Dry mouth is common among older people, affecting between 20% and 30% of those over 65 [2,3]. There are two aspects to the condition. Salivary gland hypofunction (SGH) is the state of having low salivary flow, while xerostomia refers to the subjective sensation of dry mouth. Thus, the former is a sign, and the latter a symptom (or set of symptoms). The extent to which they coincide is controversial, but the epidemiological evidence suggests that they are far from concordant [4].

By no means a trivial condition, dry mouth has a considerable impact on sufferers (Table 1). Not only is it one of the major contributors to impaired oral-health-related quality of life (OHRQoL) among adults of any age [5,6,7,8,9,10,11], but sufferers of dry mouth have difficulty eating and swallowing, halitosis, poor sleep, and considerably higher tooth decay rates [12], along with difficulty wearing dental prostheses (partial or complete dentures).

## 3. Causes of Dry Mouth

Medications are the most important risk factor for chronic dry mouth [2]. Although a small proportion of people suffer from the condition as a result of autoimmune conditions such as Sjögren’s syndrome, or as a side-effect of radiotherapy for head/neck cancer, more than 95% of the population burden of dry mouth arises as a result of medication use.

Drugs that are putative causes of dry mouth are referred to as “xerogenic”. Lists of xerogenic medications have been published [13,14,15] but are of limited utility because they are too inclusive and largely based on case reports. Medication classes for which there is sound evidence of xerogenicity include antidepressants, anticholinergics, opioids and bronchodilators (Table 2).

Other factors being equal, people taking large numbers of different medications have higher rates of dry mouth. That is, the greater the number and dosage of drugs being taken, the greater the severity of dry mouth (and other side-effects). In the NZ national survey of residential aged care, xerostomia was more common among those taking 5–9 medications and more so in those on 10+ medications, as well as in those taking antidepressants or bronchodilators [1]. Other studies have also observed differences in the occurrence of dry mouth by the total number of medications taken [16,17]. This leads into the notion of polypharmacy.

## 4. Polypharmacy

Polypharmacy is defined as the taking of five or more medications, whether they have been prescribed or purchased over the counter or online [18]. Since ageing is associated with progressively greater multi-morbidity (concurrent long-term health problems), older people tend to take a lot of medications [19]. A population-based study of community-dwelling Australians aged 70 or older reported the prevalence of polypharmacy to be 36%, with higher rates among older age groups [20]. Polypharmacy is most evident in residential aged care, where, for example, a national survey of facilities in NZ found the prevalence of polypharmacy to be 73% (and found that one in five residents were taking 10 or more medications, and no-one was taking none) [21]. Antihypertensives, analgesics and antireflux drugs were the most common types of medication.

Polypharmacy has become more common in recent decades, reflecting increased prescribing based on multiple clinical practice guidelines, each focusing on a single condition [19]. In many cases, of course, the number of medications being taken by an individual is entirely appropriate and safe, but polypharmacy is also a known risk factor for drug-related harms, falls, cognitive decline and frailty [19,22,23]. It also increases the risk of drug interactions [24], medication error [19], and the chance of important conditions going untreated [25]. Moreover, some medications taken by older people may no longer have any therapeutic or preventive effect. Health systems also suffer from the economic impacts of unnecessary prescribing and having to manage the consequent adverse events [26,27]. Recognition of the perils of polypharmacy has forged the practice of deprescribing, which is the planned and supervised process of dose reduction or stopping of medication that might be causing harm or is no longer beneficial (https://deprescribing.org/what-is-deprescribing, accessed on 20 August 2021).

## 5. Treating Dry Mouth and Reducing Unnecessary Medication Use

Dry mouth is difficult to treat. Broadly speaking, the therapeutic options are palliation (treating the symptoms), stimulation (increasing salivary gland output) or regeneration (growing new secretory tissue). The latter remains a theoretical possibility at this stage, while stimulation has had mixed outcomes. Palliative approaches can be inconsistent and unpredictable, as shown in Cochrane reviews [28,29,30], which have examined the evidence for the different therapeutic approaches. There is strong evidence for the efficacy of stimulation using systemic pilocarpine (in individually titrated doses of, typically, 2–5 mg) in treating dry mouth arising from Sjögren’s syndrome or from therapeutic radiation for head/neck cancer treatment, but those states comprise only a very small proportion of the population burden arising from dry mouth [31].

Given that the bulk of the population-attributable risk for dry mouth arises from medications (and polypharmacy in particular)—and that the evidence for stimulatory approaches to treating medication-related dry mouth is not strong—there is a need to examine alternative ways to prevent or ameliorate it by reducing the occurrence of polypharmacy in residential aged care.

Accordingly, interventions aiming to reduce the occurrence of dry mouth through medication review and deprescribing (and thereby reducing polypharmacy) would be a key strategy with important benefits, not only for older people and residential aged care facilities, but also for the wider health system. What is medication review? It involves a systematic, critical assessment of a patient’s medicines, which aims to arrive at an agreement with the patient on treatment, optimising the impact of medicines, and minimising medication-related problems and waste. An essential part of that process is medication reconciliation, the assembling of an accurate and complete inventory of all medications taken, regardless of source [32]. Medication review itself can have four levels [33] (Figure 1). (a) *Prescription review* considers the technical features of the prescription itself. (b) *Adherence support review* is undertaken with the patient present and focuses on medication-taking, particularly knowledge and adherence. (c) *Clinical review* involves the clinical notes and the patient, considering their use of medications with respect to their clinical condition. (d) *Clinical review with prescribing* is an extension of the former but including the authority to prescribe. Levels (c) and (d) closely involve the patient’s physician; overall, the focus should be on assessing the medication’s risks and benefits, and initiating deprescribing for those where the former outweigh the latter.

A useful distinction in undertaking medication review can be made between drugs being taken for control of disease and/or symptoms and those taken for preventive reasons, despite a degree of overlap [34]. In a recent investigation of medications causing dry mouth in a national survey of residential aged care in New Zealand, the types most strongly associated with dry mouth were antidepressants, corticosteroids, anticholinergics and bronchodilators [1]. These are prescribed largely for symptom control rather than disease control in older people, and so any deprescribing moves would need to be informed by considering the benefit:harm ratio and the likelihood of withdrawal reactions or disease rebound on cessation, along with patient and physician preferences [35].

In the context of dry mouth, this medication review aims to identify and eliminate specific drug categories which are known to be associated with the condition. Pharmacists can play an important role in the above process, particularly through working together with dentists, since medical practitioners may be more likely to act upon recommendations made by two practitioners from different fields.

## 6. The Need for Interventional Studies in Residential Aged Care

To date, there have been no interventional studies examining the efficacy of medication review for reducing the severity of dry mouth in residential aged care. What form could such an investigation take? The gold standard design would be a parallel randomised control study, where residents are randomly allocated to either medication review or some form of placebo consultation. There is the possibility of more nuanced comparisons, with recommendations from either a pharmacist alone or a combination of a pharmacist and a dentist. Allocation would need to be done at the level of the individual resident. Alternatively, allocation could be done at the facility level, but that would require a larger study and complicate the analyses because the unit of analysis would need to be the facility. A design which could be used in that case is the stepped wedge cluster randomised trial, treating each facility as a cluster, with no cluster exposed at first, then clusters randomly assigned to cross from the control (no intervention) to the intervention. By the end of the trial, all clusters would have been exposed to both the control and intervention conditions [36].

The outcome variables could be (a) the facility’s acceptance of deprescribing recommendations, and (b) dry mouth among residents. Dry mouth symptoms (and unstimulated salivary flow, if resources permit) would be measured at baseline (pre-intervention) and at follow-up (post-intervention). Given the chronic nature of dry mouth and the exposure to the medications responsible, the length of time between the intervention and the follow-up assessment would need to be six months, to allow for (i) adequate tapering and “wash-out” of the responsible medications, and (ii) recovery of the salivary system. For measuring xerostomia symptoms, an established and validated scale such as the Summated Xerostomia Inventory (SXI) would be useful [37], because it assigns participants to a place on a continuum of dry mouth severity, enabling the computation of an effect size for the observed change in symptoms arising from the intervention. Given that the minimally important difference (MID) for the SXI is 4, the proportion improving by the MID in the intervention and control groups would be able to be compared. Similarly, unstimulated flow rates pre- and post-intervention could be compared, along with effect sizes.

Such a design is, of course, not the only one which could be used to determine the efficacy of medication review and deprescribing in treating dry mouth, and it would be worth considering and investigating the feasibility of other approaches. It is hoped that this review raises awareness of the issue of dry mouth in residential aged care and provides some impetus towards the development of multidisciplinary collaborations between pharmacists and dentists, which aim to ameliorate and further investigate the problem.

## 7. Conclusions

Pharmacists and health practitioners are likely to be aware of the role of polypharmacy in falls and confusion, but they may not be cognisant of drug-induced dry mouth, its significance for oral health, and its negative impact on quality of life. Thus, there is a need for pharmacy and oral health personnel to work more closely—along with general practitioners, geriatricians and nurses—to raise awareness of xerostomia and ensure that sufferers are appropriately managed and advised. Practitioners should routinely enquire about dry mouth when assessing residents, and specific enquiries about dry mouth should be included in the standard medical histories for new admissions. Given that the major risk factor for dry mouth is polypharmacy, pharmacists, doctors and dentists should work together to prevent this from occurring, or at least reduce its severity. Key strategies are medication review and deprescribing. There is also a need for interventional studies to determine the efficacy of those strategies in reducing the occurrence of dry mouth.

## Figures and Tables

**Figure 1 pharmacy-09-00162-f001:**
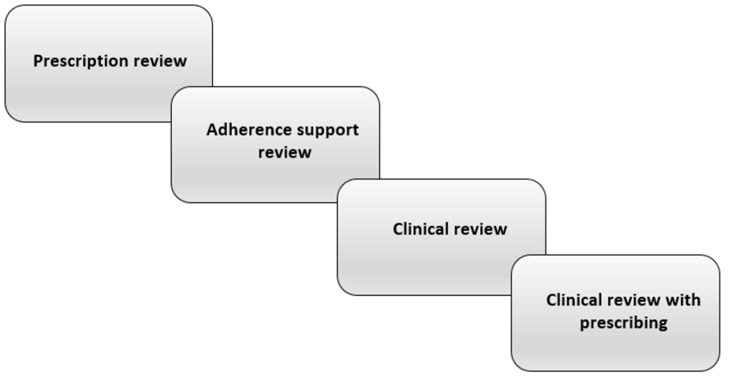
The levels of pharmacist-led medication review [33].

**Table 1 pharmacy-09-00162-t001:** Overview of dry mouth’s impacts on sufferers.

Physical and Functional Impacts	Psychosocial Impacts
Difficulty eating/swallowing	Symptoms of dry mouth
More tooth decay	Compromised quality of life
Problems with dentures	Halitosis
Infections—salivary glands, mucosa	Poor sleep
Compromised taste sensation	Distress

**Table 2 pharmacy-09-00162-t002:** The main medication classes known to be associated with dry mouth (adapted from Villa et al. [4]).

Medication Type	Mechanism of Action
Gastrointestinal agents e.g., Hyoscine, hyoscyamine, belladonna alkaloids, atropine	Block muscarinic receptors
Antiemetics e.g., prochlorperazine,	Block dopamine D2, serotonin types 2–4, histamine type 1 and acetylcholine receptors
Appetite suppressants/stimulants e.g., Phentermine, sibutramine	Inhibit CNS uptake of norepinephrine, serotonin and dopamine
Cardiovascular agents e.g., Atenolol, metoprolol, prazosin, clonidine	Block α_1_- and β_2_-adrenergic receptors
Urological e.g., Oxybutynin, propantheline, darifenacin, solifenacin, tolterodine, mirabegron	Block muscarinic receptors and α_1_-adrenergic receptors
Muscle relaxants Cyclobenzaprine, orphenadrine	Act as α_1_-adrenergic receptor agonists, and H2 histamine blockers
Analgesics e.g., Opioids, tramadol, gabapentin, pregabalin.	Block noradrenaline reuptake in the CNS and so inhibit the salivary reflex arc
Anticonvulsants e.g., Carbamazepine	Act centrally to reduce neurotransmitter release
Sedatives—benzodiazepines & Z-drugs e.g., Zolpidem, zopiclone	Enhance GABA effect in CNS, reduce the salivary secretory reflex, and block muscarinic, α_1_- and β_2_-adrenergic receptors
Antipsychotics e.g., Olanzapine, clozapine, amisulpiride	Block neurotransmitter uptake (various)
Antidepressants e.g., Tricyclics (e.g., amitriptyline), SSRIs and SNRIs	Anticholinergic; increase serototinn and noradrenaline at the synaptic cleft.
Bronchodilators e.g., Ipratropium, tiotropium, salbutamol, salmeterol, eformoterol, umeclidinium	2 types: β agonists and antimuscarininc, Block muscarinic receptors M1 and M3,
Antihistamines- sedating only e.g., Diphenhydramine, doxylamine, chlorpheniramine, promethazine	Central inhibitory action on histamine type 1 and muscarinic receptors
CNS Stimulants e.g., Caffeine, pseudoephedrine, amphetamines	α_1_ and α_2_ agonists.

## Data Availability

Not applicable.

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
