# Peer review of "The Challenge of Medication-Induced Dry Mouth in Residential Aged Care"

_pharmacy, 2021, doi:10.3390/pharmacy9040162_

Round 1

Reviewer 1 Report

Specific Comments

Occurrence and impact of dry moth

  • OK

Causes of dry mouth

  • Given this is a review of dry mouth I am surprised there is not a table of medicines associated with dry mouth to educate the reader. The authors draw on papers from the 1980s/1990s as being not useful. The authors then state that medications classes which “there is sound evidence of xerogenicity include antidepressants, anticholinergics, opioids and bronchodilators”. This statement is unreferenced. Given this paper centres on a review on the challenged of medicine-induced dry mouth I expected a greater discussion in this area. A table would be helpful. E.g. medicines (categorised into strength of evidence) which are anticholinergic and likely to cause dry mouth – do eye drops or only oral medication.
  • I am surprised the paper by Tan et al (https://pubmed.ncbi.nlm.nih.gov/29071719/) has not been included in this review as it focusses on older people.

Polypharmacy

  • I agree that polypharmacy is a concern, and it has become a buzz word over the last two decades. Furthermore, deprescribing has become quite popular too! And for good reasons too. However, polypharmacy is also a risk factor for undertreatment in some patients (e.g. consider need to prescribe a laxative when prescribing opioids). I think it is worth balancing polypharmacy with reference to the paper by Dutch researchers et al in 2008 (Br J Clin Pharmacol.2008 Jan; 65(1): 130–133.). By adding this in it makes sure the reader pauses to consider the issues
  • Page 3 line 89 “ahs’ should be ‘has”

Treating dry mouth

  • I found this section quite weak. A reader looking for a review on the topic is likely to want to see some practical solutions on the management of dry mouth. e.g. using cholinergic mouth drops? Using artificial saliva? Do they work? is there evidence? Contraindications to treatments Etc. I realise that this paper is looking at aged care, but I think to be of benefit to readers in the absence of hard data there needs to be some discussion on some specific treatments even if the evidence base is weak and based only on anecdotal or experience.

Reducing polypharmacy in residential aged care

  • I am not familiar with the four levels of medication review. It is not referenced. A prescription review can be done by another health professional not just a pharmacist. A treatment review can be done between different clinicians excluding a pharmacist. For example, in aged care it is not uncommon for a nurse and Dr to review a resident’s medicines and then refer for a medication review. I think these definitions either need referencing or perhaps refer to commonly accepted definitions of medication review. Figure 1 will then need changing.
  • Page 4 line 141: “steroids” do you mean corticosteroids or anabolic steroids ?

The need for interventional studies in residential aged care

  • Isn’t there a need to incorporate some assessment of medicines to help with dry mouth. i.e. deprescribing some medications is one element, but there are instances where deprescribing isn’t possible or feasible. E.g. corticosteroids may not be able to be reduced/stopped.

Author Response

Causes of dry mouth

Given this is a review of dry mouth I am surprised there is not a table of medicines associated with dry mouth to educate the reader. The authors draw on papers from the 1980s/1990s as being not useful. The authors then state that medications classes which “there is sound evidence of xerogenicity include antidepressants, anticholinergics, opioids and bronchodilators”. This statement is unreferenced. Given this paper centres on a review on the challenged of medicine-induced dry mouth I expected a greater discussion in this area. A table would be helpful. E.g. medicines (categorised into strength of evidence) which are anticholinergic and likely to cause dry mouth – do eye drops or only oral medication.

I am surprised the paper by Tan et al (https://pubmed.ncbi.nlm.nih.gov/29071719/) has not been included in this review as it focusses on older people.

AU response: corrected. Re the Tan et al paper, there are many papers focusing on older people and medications, and we cannot include them all, but we have now cited this one.

Polypharmacy

I agree that polypharmacy is a concern, and it has become a buzz word over the last two decades. Furthermore, deprescribing has become quite popular too! And for good reasons too. However, polypharmacy is also a risk factor for undertreatment in some patients (e.g. consider need to prescribe a laxative when prescribing opioids). I think it is worth balancing polypharmacy with reference to the paper by Dutch researchers et al in 2008 (Br J Clin Pharmacol.2008 Jan; 65(1): 130–133.). By adding this in it makes sure the reader pauses to consider the issues.

AU response: Thank you for directing our attention to the excellent paper by Kuijpers et al – we found that useful and have worked it into the text of our paper (in Section 4 – now renumbered, having previously been Section 3).

Page 3 line 89 “ahs’ should be ‘has”

AU response: corrected.

Treating dry mouth

I found this section quite weak. A reader looking for a review on the topic is likely to want to see some practical solutions on the management of dry mouth. e.g. using cholinergic mouth drops? Using artificial saliva? Do they work? is there evidence? Contraindications to treatments Etc. I realise that this paper is looking at aged care, but I think to be of benefit to readers in the absence of hard data there needs to be some discussion on some specific treatments even if the evidence base is weak and based only on anecdotal or experience.

AU response: Treating dry mouth is difficult and a whole new paper in itself, because the various causes of dry mouth must first be elucidated before its treatment can be considered. This is because different approaches are required and these differ in their evidence bases. The topic has been the focus of a number of Cochrane reviews in recent years. We have now cited those, and we have endeavoured to make that section more clear.

Reducing polypharmacy in residential aged care

I am not familiar with the four levels of medication review. It is not referenced. A prescription review can be done by another health professional not just a pharmacist. A treatment review can be done between different clinicians excluding a pharmacist. For example, in aged care it is not uncommon for a nurse and Dr to review a resident’s medicines and then refer for a medication review. I think these definitions either need referencing or perhaps refer to commonly accepted definitions of medication review. Figure 1 will then need changing.

AU response: we have rewritten the medication review section.

Page 4 line 141: “steroids” do you mean corticosteroids or anabolic steroids ?

AU response: corticosteroids – we have now made this clear.

The need for interventional studies in residential aged care

Isn’t there a need to incorporate some assessment of medicines to help with dry mouth. i.e. deprescribing some medications is one element, but there are instances where deprescribing isn’t possible or feasible. E.g. corticosteroids may not be able to be reduced/stopped.

AU response: corrected.

Reviewer 2 Report

Thank you for the submission.  Although not particularly novel, it is quite interesting and reasonably well-written.  I thought that more information could have been included on the impact of dry mouth and poor oral health in the elderly, including preventable hospitalisations. I am unsure of the origin of the 4 levels of medication reviews (the figure doesn’t have a reference). More typically, there are three levels of pharmacist-led medication reviews (see Hatah E, Braund R, Tordoff J, Duffull SB. A systematic review and meta-analysis of pharmacist-led fee-for-services medication review. Brit J Clin Pharmacol. 2014;77:102-15). Level one review is an assessment of technical components of a prescription. Level two includes adherence support and patient participation to improve medication taking behaviour (as per the MURs funded in some countries e.g. UK, Australia). These are typically performed in the pharmacy. Level three is a much more comprehensive assessment of medication use, in collaboration with other health professionals and prescribers involved in patient care (and typically performed in aged care settings or patients’ homes). In Australia, as an example, there are government-funded clinical medication review (level 3 review) services for the aged care setting (residential medication management review (RMMR) service) and for individuals living at home (home medicines review (HMR)). International pharmacist-led medication review (level three) services, which meet the definition of clinical medication review, include Medication Therapy Management in the United States, and MedsCheck Long-Term Care in Canada.  In fact, the figure is unnecessary. It would be more useful to have a table of the drug groups commonly associated with dry mouth and the mechanisms involved. Is there more evidence that polypharmacy is associate with dry mouth? The single reference provided is not particularly robust (dichotomising by less/greater than 10 medicines).

There needs to be more information on the actual treatment of dry mouth, which is very superficial at present. There is too much information on polypharmacy and trial designs (it is not particularly useful or necessary to get into stepped wedge cluster RCTs etc). Again, more information on the actual topic (dry mouth, specifically medication-induced) would be better. The balance is not right, as is.

Author Response

I am unsure of the origin of the 4 levels of medication reviews (the figure doesn’t have a reference). More typically, there are three levels of pharmacist-led medication reviews (see Hatah E, Braund R, Tordoff J, Duffull SB. A systematic review and meta-analysis of pharmacist-led fee-for-services medication review. Brit J Clin Pharmacol. 2014;77:102-15). Level one review is an assessment of technical components of a prescription. Level two includes adherence support and patient participation to improve medication taking behaviour (as per the MURs funded in some countries e.g. UK, Australia). These are typically performed in the pharmacy. Level three is a much more comprehensive assessment of medication use, in collaboration with other health professionals and prescribers involved in patient care (and typically performed in aged care settings or patients’ homes). In Australia, as an example, there are government-funded clinical medication review (level 3 review) services for the aged care setting (residential medication management review (RMMR) service) and for individuals living at home (home medicines review (HMR)). International pharmacist-led medication review (level three) services, which meet the definition of clinical medication review, include Medication Therapy Management in the United States, and MedsCheck Long-Term Care in Canada. 

AU response: We originally used the 4-level schema proposed by Geurts et al (2012). Following this reviewer’s feedback, we have adopted the 4-step one of Hatah et al (2014).

In fact, the figure is unnecessary. It would be more useful to have a table of the drug groups commonly associated with dry mouth and the mechanisms involved. Is there more evidence that polypharmacy is associate with dry mouth? The single reference provided is not particularly robust (dichotomising by less/greater than 10 medicines).

AU response: Our original submission had no Figure, but then we were informed by the publisher that we had to have at least one Figure and one Table. Hence, we hastily assembled the Figure. We would prefer to not have it, but there is a requirement to have one!  Re the issue of a Table of the drug groups commonly associated with dry mouth, we have now added this. As for the single reference on polypharmacy and aspects of dry mouth, the paper we used was chosen because those nursing home data are nationally representative, which is a rare thing indeed in this area of research. we have now added more – studies by Smidt et al (2010) and Johanson et al (2015).

There needs to be more information on the actual treatment of dry mouth, which is very superficial at present. There is too much information on polypharmacy and trial designs (it is not particularly useful or necessary to get into stepped wedge cluster RCTs etc). Again, more information on the actual topic (dry mouth, specifically medication-induced) would be better. The balance is not right, as is.

AU response: As pointed out above, the treatment of dry mouth is difficult. We have endeavoured to make that section more informative and to better highlight the challenges.

Reviewer 3 Report

The theme proposed for the manuscript is quite interesting, however, I was expecting to see it developed in a deeper way. It is not clear what the objective of the work is. Please write a sentence with a clear objective. This is a descriptive review study, but the authors need to better demonstrate what the importance of the manuscript is to the readers and what methodology was used in its preparation. The authors dedicate a point to polymedication, but I do not think that this will be an objective of the work and end up repeating in this point some information that they had described in point 2 "causes of dry mouth". Section 2 needs to be more in-depth as it seems to me that it will be an important point for the readers. Figure 1 as it stands is not very interesting, as it describes in a general way the process of medication review already described in several other manuscripts. the authors could adapt it to the specific case of the objective of the study, which could be of interest for clinical practice.

Author Response

The theme proposed for the manuscript is quite interesting, however, I was expecting to see it developed in a deeper way. It is not clear what the objective of the work is. Please write a sentence with a clear objective.

AU response: We are in some ways hampered here by the structure required by the journal. However, we have amended this by inserting an Introduction section (numbered 1 – which has, of course, altered the numbering of the other sections) and hope that the objective of the paper is clearer now.

This is a descriptive review study, but the authors need to better demonstrate what the importance of the manuscript is to the readers and what methodology was used in its preparation. The authors dedicate a point to polymedication, but I do not think that this will be an objective of the work and end up repeating in this point some information that they had described in point 2 "causes of dry mouth".

AU response: This is a narrative review which deals with an important aspect of oral health in residential aged care in order to raise awareness of those particular issues for pharmacists.

Section 2 needs to be more in-depth as it seems to me that it will be an important point for the readers. Figure 1 as it stands is not very interesting, as it describes in a general way the process of medication review already described in several other manuscripts. the authors could adapt it to the specific case of the objective of the study, which could be of interest for clinical practice.

AU response:  We have endeavoured to make Section 2 (now 3) more informative for readers.

Round 2

Reviewer 1 Report

Thank you for addressing all of my comments. The changes have improved the manuscript. I would still like to see an expanded section on the treatment to make this paper more useful for readers, particularly practising pharmacists. What strength of pilocarpine? how long does it take for it to work? What level of evidence supports it use or what is the NNT? Any medications in development. It doesnt matter if the evidence is weak as long as this is stated. I would expect to see more on the medical treatment of this condition in a review paper.

Author Response

In respect of pilocarpine, its use by oral medicine specialists in treating dry mouth is very much tailored to the individual patient, with careful titration required so that the more unpleasant parasympathomimetic side-effects are minimised. Accordingly, we are reluctant to recommend a specific dose or NNT, etc. We are not aware of any medications in development. The overall topic of the pharmacologic treatment of dry mouth—and newer pharmacological therapy approaches—is one which requires its own review paper, given the challenges. Our paper is focused much more on the problem of medication-induced dry mouth in residential aged care; this is appropriate because of the fact that it comprises the far greater proportion of cases of dry mouth in older people.

Reviewer 3 Report

Regarding the section on polypharmacy and medication review, I think that instead of such general information, it could be more focused on the topic under discussion "medication-induced dry mouth in residential aged care".

Author Response

AU response: We have now combined Sections 5 and 6 to better reflect their content and part in the idea sequence in the paper:

Section 1  - Introduction to the paper

Section 2 – Dry mouth is common and adveresly affects sufferers

Section 3 – Medications are by far the most important cause of that impact

Section 4 – Polypharmacy is common in older people and especially in aged care

Section 5 – Dry mouth is difficult to treat, and polypharmacy is a challenge

Section 6 – Interventional studies in age care are needed to determine whether
                 medication review would reduce the problem of dry mouth

Round 3

Reviewer 1 Report

This paper needs some expansion on treatment of dry mouth for it to be of interest to readers in pharmacy. eg. what % pilocarpine ise used eg 0.3%. Perhaps they can provide some general guidance to the readers.

eg. https://jphcs.biomedcentral.com/articles/10.1186/s40780-018-0099-x

https://www.jamda.com/article/S1525-8610(20)30559-4/fulltext 

Author Response

As we pointed out in the paper, in the rare cases where clinicians might try pilocarpine with medication-induced dry mouth sufferers, the dose is titrated to the individual patient, so there are no hard-and-fast rules for dosage. In any case, resorting to pilocarpine without first considering medication review and deprescribing would be (ironically) merely compounding the patient’s polypharmacy. The point of our paper is to argue (a) that medications are responsible for the great majority of dry mouth out there, and (b) that the answer to the problem is to reduce patients’ exposure to those rather than adding another one (particularly one which can have unpleasant side-effects). Nevertheless, we have now mentioned dosage in line 106.

Of the two examples above, the first relates to Sjögren’s syndrome (which is essentially outside our paper’s scope) and the other is an under-powered (N = 10) pilot study with older people described in what is essentially a letter to the editor.